# Nanosized Cu-SSZ-13 and Its Application in NH₃-SCR

**Ana Palčić [1]**, **Paolo Cleto Bruzzese [2]**, **Kamila Pyra [3]**, **Marko Bertmer [2]**, **Kinga Góra-Marek [3]**, **David Poppitz [4]**, **Andreas Pöppl [2]**, **Roger Gläser [4]** and **Magdalena Jabłońska [4],***

[1] Laboratory for the Synthesis of New Materials, Division of Materials Chemistry, Ruđer Bošković Institute, Bijenička 54, 10000 Zagreb, Croatia; ana.palcic@irb.hr

[2] Felix Bloch Institute for Solid State Physics, Universität Leipzig, Linnéstr. 5, 04103 Leipzig, Germany; paolo.bruzzese@physik.uni-leipzig.de (P.C.B.); bertmer@physik.uni-leipzig.de (M.B.); poeppl@physik.uni-leipzig.de (A.P.)

[3] Faculty of Chemistry, Jagiellonian University in Kraków, Gronostajowa 2, 30-387 Kraków, Poland; kamila.pyra@doctoral.uj.edu.pl (K.P.); kinga.gora-marek@uj.edu.pl (K.G.-M.)

[4] Institute of Chemical Technology, Universität Leipzig, Linnéstr. 3, 04103 Leipzig, Germany; david.poppitz@uni-leipzig.de (D.P.); roger.glaeser@uni-leipzig.de (R.G.)

\* Correspondence: magdalena.jablonska@uni-leipzig.de; Tel.: +49-0341-97-36303

**Abstract:** Nanosized SSZ-13 was synthesized hydrothermally by applying N,N,N-trimethyl-1-adamantammonium hydroxide (TMAdaOH) as a structure-directing agent. In the next step, the quantity of TMAdaOH in the initial synthesis mixture of SSZ-13 was reduced by half. Furthermore, we varied the sodium hydroxide concentration. After ion-exchange with copper ions (Cu²⁺ and Cu⁺), the Cu-SSZ-13 catalysts were characterized to explore their framework composition (XRD, solid-state NMR, ICP-OES), texture (N₂-sorption, SEM) and acid/redox properties (FT-IR, TPR-H₂, DR UV-Vis, EPR). Finally, the materials were tested in the selective catalytic reduction of $NO_x$ with ammonia (NH₃-SCR). The main difference between the Cu-SSZ-13 catalysts was the number of Cu²⁺ in the double six-membered ring (6MRs). Such copper species contribute to a high NH₃-SCR activity. Nevertheless, all materials show comparable activity in NH₃-SCR up to 350 °C. Above 350 °C, NO conversion decreased for Cu-SSZ-13(2–4) due to side reaction of NH₃ oxidation.

**Keywords:** chabazite; SSZ-13; preparation method; copper species; IR spectroscopy; NH₃-SCR

## 1. Introduction

SSZ-13 and SAPO-34 are typical examples of zeolite materials with a chabazite (CHA) structure. SSZ-13 is an aluminosilicate CHA-type material with high Si content and SAPO-34 is its silicoaluminophosphate counterpart. The CHA framework is composed of four-, six-, and eight-membered rings arranged to form a tridimensional system of channels perpendicular to each other (0.38 × 0.38 nm; R3m (#166) space group) [1]. Its structure can be represented as *cha* cages connected via double six-membered rings and thus, is a member of the *abc*-six frameworks family. The incorporation of Cu ions into the CHA zeolite is a key factor in the preparation of the Cu-CHA catalysts. Techniques such as ion-exchange [2], chemical vapor deposition [3], solid-state ion-exchange (SSIE) [4], one-pot preparation with Cu ions introduced by the structure-directing agent [5], and wet mixing and impregnation [6] have been examined for the preparation of Cu-CHA materials. The most frequently applied ion-exchange method with aqueous copper salt solution allows more Cu²⁺ ions to be located at the exchange sites of SSZ-13. In Cu-SSZ-13, copper species are predominantly present as isolated Cu²⁺ ions, even at higher ion-exchange degrees [2]. There are four cationic sites in the CHA framework [7]: site I—located at the six-membered ring in the

elliptical cavity; site II—placed near the center of the elliptical cavity; site III—located in the center of the hexagonal prism face; site IV—near to the eight-membered ring window. It was observed that the state of the copper species could be strongly affected by the preparation method. Yan et al. [6] found that the isolated $Cu^{2+}$ species are mainly located in the elliptical cavity (site I) for the ion-exchanged Cu-SAPO-34 (silicoaluminophosphate zeotype isostructural to SSZ-13). $Cu^{2+}$ located at the site I was more stable than that located at the site IV. Deka et al. [3] and Xue et al. [7] further investigated the materials obtained through the ion-exchange method. Their results confirmed that isolated $Cu^{2+}$ species associated with the six-membered ring window, and placed into the ellipsoidal cavity of SAPO-34 were the active sites for $NH_3$-SCR. This statement can also be applied to Cu-SSZ-13 [7–9]. Such catalysts show excellent NO conversion and $N_2$ selectivity in a wide temperature window (250–550 °C) in comparison to medium pore zeolites (e.g., Cu-ZSM-5) [10,11].

Furthermore, the size of the CHA crystals might affect the efficiency of the ion-exchange process, i.e., the quantity of the initial cations, which are exchanged with the needed cations as the surface area available for direct contact with the solution is determined by the size of the particles. In addition, depending on the content of copper in Cu-SSZ-13, copper ions can occupy different positions within the crystal structure. At a lower Cu content, they are mainly positioned at sites in the six-membered rings and at higher copper content, within the large cages [12]. Overall, the preparation of the samples with a predominantly single type of Cu species is not straightforward. Additionally, the relatively high cost of the most frequent and efficient organic structure-directing agent (OSDA) N,N,N-trimethyl-1-1-adamantammonium hydroxide (TMAdaOH) to prepare CHA-type materials strongly limits wide applications of Cu-SSZ-13. To date, researchers have concentrated on the ways of cost reduction of the Cu-SSZ-13 catalysts [13,14]. In general, high quantities of OSDA are necessary to prepare nanosized crystals (<500 nm) of high-silica zeolite materials [15]. For this reason, any diminution of an OSDA amount in the synthesis mixture, which at the same time, enables to obtain a similar effect, i.e., prepare materials of similar properties, is highly advantageous. The benefits of decreasing the size of zeolite crystals are shorter intra-crystal channels and larger external surface area and thus, consequently a shorter diffusion path as well as more exposed and available active sites. Gao et al. [16] first raised that for the Cu-SSZ-13 catalysts the reaction kinetics are controlled by intra-particle diffusion limitations. They observed a decrease in turnover frequency (TOF) for Cu-SSZ-13 with different copper loadings (1.31–5.15 wt. %). Furthermore, Wang et al. [17] measured the $NH_3$-SCR rates on (1–2 wt. %)Cu-SSZ-13 with various particle size (0.4–2 μm), achieved by modulating seed crystals and pH in the synthesis media prior to crystallization. The authors concluded that the $NH_3$-SCR reaction was free of intra-crystalline diffusion because of the identical TOF. Bates et al. [18] found that the $NH_3$-SCR rates on Cu-SSZ-13 increased linearly with Cu loading (n(Cu)/n(Al) = 0.04–0.2) and concluded that the reaction kinetics is not limited by intra-crystalline diffusion. Otherwise, the effect of particle size (less than 0.5 μm) of Cu-SSZ-13 on its catalytic properties in $NH_3$-SCR was less intensively investigated [19].

Thus, within the frame of this investigation, a series of SSZ-13 samples with varying particle sizes (mean values in the range of 216–288 nm) were synthesized. For practical applications, the Cu-SSZ-13 materials were prepared and characterized to explore their structure and texture as well as their acidic and redox properties. The Cu-SSZ-13 materials were characterized by X-ray diffraction (XRD), solid-state nuclear magnetic resonance (NMR) spectroscopy, inductively coupled plasma optical emission spectrometry (ICP-OES), $N_2$-sorption, scanning electron microscopy (SEM) with energy dispersive X-ray (EDX) analysis, Fourier transform infrared (FT-IR) spectroscopy, temperature-programmed reduction (TPR) with $H_2$, diffuse reflectance UV-Vis (DR UV-Vis) spectroscopy and electron paramagnetic resonance (EPR) spectroscopy. The catalytic properties of the prepared Cu-SSZ-13 were evaluated in $NH_3$-SCR.

## 2. Results and Discussion

### 2.1. Structural and Textural Properties, Morphology of Cu-SSZ-13

Table 1 presents the prepared materials and their compositions. Briefly, SSZ-13(1) was synthesized hydrothermally by applying TMAdaOH as a structure-directing agent. SSZ-13(2) contains the quantity of TMAdaOH reduced by half compared to the initial synthesis mixture of SSZ-13(1). In the following materials—SSZ-13(3, 4) we varied the sodium hydroxide concentration. The XRD patterns (Figure 1a) of the solid final products—prepared from the studied reaction mixtures hydrothermally treated at 150 °C for a certain period, correspond to the CHA-type materials. No impurity phase was observed. The narrow peaks suggest the high crystallinity degree of the samples. Thus, the changes in the chemical composition of the reaction mixture did not affect the phase composition of the final product. Figure 1b shows the XRD patterns of Cu-exchanged SSZ-13, indicating that the chabazite (CHA) structure was maintained during the ion-exchange. $Cu_2O$ and $CuO$ phases at ca. $2\theta = 35.29, 36.30,$ $38.49$ and $38.72°$ [20,21] are not observed. The relatively low content of copper (4.4–5.2 wt. %) therefore can result in either a high copper dispersion or the amorphous nature of $Cu_2O$ and $CuO$.

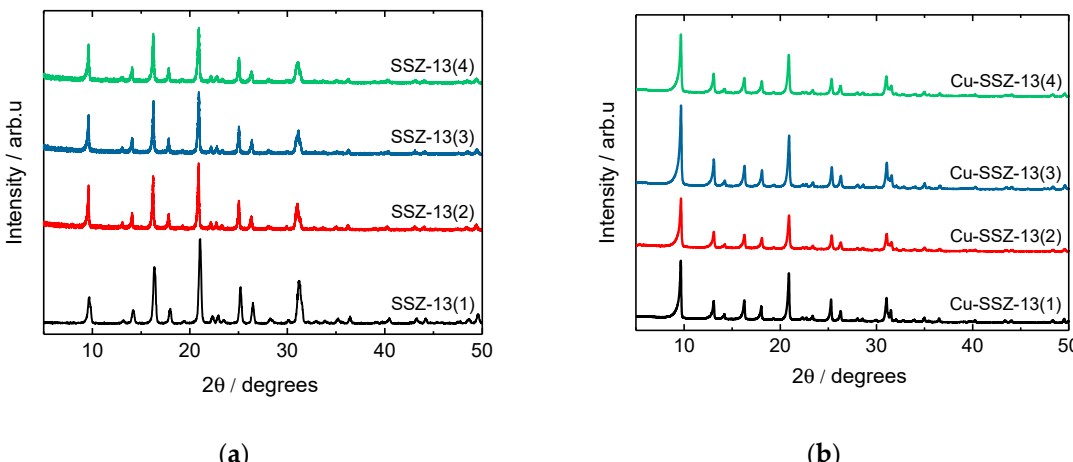

(**a**)                                         (**b**)

**Figure 1.** (**a**) XRD patterns of the products obtained from the synthesis mixtures hydrothermally treated at 150 °C, (**b**) XRD patterns of Cu-exchanged SSZ-13 (sample labels as in Table 1).

**Table 1.** Molar oxide ratio of the studied synthesis mixtures, the time of hydrothermal treatment at 150 °C and the phase composition of the end product.

| Sample | Molar Batch Composition | Time/d | Product |
|--------|-------------------------|--------|---------|
| SSZ-13(1) | 1 $SiO_2$:0.0167 $Al_2O_3$:0.25 NaOH:0.25 TMAdaOH:15 $H_2O$ | 6 | CHA |
| SSZ-13(2) | 1 $SiO_2$:0.0167 $Al_2O_3$:0.25 NaOH:0.125 TMAdaOH:15 $H_2O$ | 6 | CHA |
| SSZ-13(3) | 1 $SiO_2$:0.0167 $Al_2O_3$:0.2 NaOH:0.125 TMAdaOH:15 $H_2O$ | 10 | CHA |
| SSZ-13(4) | 1 $SiO_2$:0.0167 $Al_2O_3$:0.3 NaOH:0.125 TMAdaOH:15 $H_2O$ | 5 | CHA |

The SEM images (Figure 2) of the initial sample with the highest TMAdaOH concentration show that the crystals have cubic morphology and are rather uniform in both size and shape. However, reducing the TMAdaOH content results in crystals having more rounded edges and smoothed surfaces. Furthermore, an intergrowth of the crystals and an apparently broader crystal size distribution were observed. The crystal size varies from 80 to 600 nm and a tendency towards smaller crystal sizes with broader crystal size distribution by reducing TMAdaOH content was found. The 100-line measurements in corresponding SEM image result in mean values: 288 nm—Cu-SSZ-13(1), 250 nm—Cu-SSZ-13(2), 266 nm—Cu-SSZ-13(3), 216 nm—Cu-SSZ-13(4). Clearly, the presence of higher quantity of sodium in the systems of Cu-SSZ-13(2–4) influences the nuclei formation within a longer reaction period, i.e., nuclei formation and crystal growth occurred during a longer period, resulting in crystals with

a smaller size and a broader size distribution. Moreover, it affects the crystal growth process and leads to crystals that are more spherical. Finally, the EDX analysis (10 positions on each sample) revealed a homogenous distribution of the elements, i.e., any particles that might correspond to isolated copper-containing material were observed. The data are not shown here due to the better accuracy of ICP-OES (bulk analysis).

The data on the elemental composition of the Cu-CHA materials obtained by ICP-OES (Table 2) show the n(Si)/n(Al) ratio in all samples studied ranging from 13.9 up to 17.8 with its average value being n(Si)/n(Al) = 15.6. This suggests that in the studied samples, the reduction of the concentration of the organic structure-directing agent, TMAdaOH, in the reaction mixture does not affect the framework composition of the final material. Hence, the CHA-type zeolite synthesis approaches demonstrated in this work represent more economical and environmentally friendly routes than CHA synthesis from the initial synthesis mixture (using double amount of TMAdaOH). Namely, in general, the organic structure-directing agents are added to the systems to increase the framework n(Si)/n(Al) ratio. Herein, a similar framework composition was achieved with half of the amount of TMAdaOH, which is a costly chemical and its production requires usage of organic solvents.

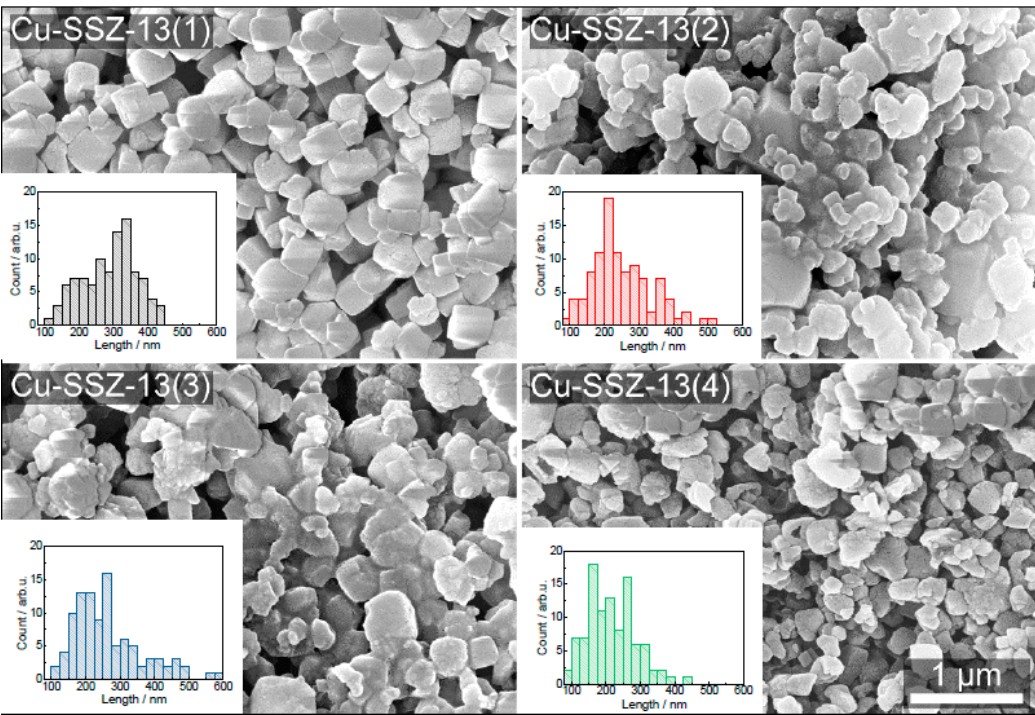

**Figure 2.** SEM images of Cu-exchanged SSZ-13. The crystallite size distribution was constructed by counting 100 crystallites randomly from SEM images.

**Table 2.** The results of elemental analysis of the Cu-exchanged chabazite samples, their textural properties (specific surface area: $a_S$(BET); micropore and total pore volumes: $V_{MIC}$ and $V_{TOT}$) determined from the $N_2$-sorption isotherms and the $H_2$ consumption during TPR.

| Sample | Si; Al; Cu Content/wt. % | $n$(Si)/$n$(Al); $n$(Cu)/$n$(Al) | $a_S$(BET)/m$^2$ g$^{-1}$ | $V_{MIC}$/cm$^3$ g$^{-1}$ | $V_{TOT}$/cm$^3$ g$^{-1}$ | $H_2$ Consumption /mmol $H_2$ g$^{-1}$ |
|---|---|---|---|---|---|---|
| Cu-SSZ-13(1) | 37.6; 2.6; 4.4 | 13.9; 0.72 | 575 | 0.268 | 0.314 | 1.08 |
| Cu-SSZ-13(2) | 38.6; 2.2; 4.9 | 16.9; 0.95 | 582 | 0.265 | 0.352 | 1.16 |
| Cu-SSZ-13(3) | 37.0; 2.0; 5.2 | 17.8; 1.10 | 615 | 0.275 | 0.442 | 1.25 |
| Cu-SSZ-13(4) | 34.6; 2.4; 5.1 | 13.8; 0.90 | 591 | 0.262 | 0.379 | 1.18 |

The chemical composition of the prepared materials is similar (n(Si)/n(Al) ≈ 14–18), however, the differences in the distribution in the framework Al can affect the acid properties, i.e., type,

concentration and strength of acid sites. Namely, CHA-type zeolite materials, which were synthesized in the presence of solely TMAdaOH, were found to contain only isolated framework Al [22]. The addition of $Na^+$ to the zeolite preparation mixture, maintaining the concentration of cations and other preparation parameters, yielded crystalline SSZ-13 zeolites of a fixed n(Si)/n(Al) ratio, but with paired Al sites (sequence Al–O(–Si–O)$_2$–Al, based on MAS-NMR). The amount of the paired Al sites has been found as a linear function of the $Na^+$ incorporated into the crystalline solids [23]. Al distribution within the zeolite framework determines the ability of the zeolite for ion-exchange with divalent cations. DFT calculations demonstrated that $Cu^{2+}$ prefers to site at 6MRs possessing paired Al sites [24]. As aforementioned, the acidic properties of the materials were studied by $NH_3$ sorption experiments followed by IR spectroscopy according to the well-known methodology described elsewhere [25,26]. The Brønsted and Lewis acid sites concentrations in the Cu-SSZ-13 zeolites determined by IR quantitative measurements are given in Table 3. A reduction of the Brønsted acid sites number is clearly visible for material with the quantity of TMAdaOH reduced by half, Cu-SSZ-13(2). This decline in the density of protonic sites is accompanied, however, by an increase in Lewis acid sites density. In addition to Al-atoms, the exchangeable copper cations can also bond coordinatively ammonia molecules providing a contribution to the total density of Lewis acid sites detected with this probe. As above, an increase in the concentration of Lewis sites is easily explained by the higher Cu content for Cu-SSZ-13(2) than Cu-SSZ-13(1) (4.9 *versus* 4.4 wt. %, respectively, Table 2). For the Cu-SSZ-13(3,4) materials prepared with varied sodium hydroxide concentration (0.2–0.3 NaOH), the Brønsted acidity appeared in minor extent. This can be explained by either the higher n(Cu)/n(Al) ratio of those zeolites or the various forms of copper sites; the latter is controlled by Al atoms' location in the zeolite framework. The Al atoms siting (Al pairs or single Al atoms) can be considered as the factor for stabilization of $Cu^{2+}$, $Cu^+$ and $Cu_{oxo}$ (e.g., $[Cu–O–Cu]^{2+}$) species. Most probably, in Cu-SSZ-13(3,4), the hydrolysis leading to oxide-like species formation is more pronounced, thus the Al framework atoms are balanced by protons, not by exchangeable copper cations. Finally, in Cu-SSZ-13(3,4) the co-existence of ion-exchange copper cations and small oligomeric or bulky CuO species is assumed.

Figure 3 shows the $^{27}Al$ NMR spectra for the H-SSZ-13 samples. The majority of Al is tetrahedrally coordinated and fully incorporated in the framework, as indicated by the peak at 59 ppm for Cu-SSZ-13(1) and 60 ppm for Cu-SSZ-13(2–4). Although Al is in tetrahedral coordination in zeolite framework, its local environment differs among the samples. However, additional experiments are required to make a more detailed analysis and consequently, precisely designate the Al siting. Those $Al^{4+}$ in Si(OH)Al units can provide Brønsted acidity. Only a small amount of extra-framework octahedral Al at 0 ppm is noticeable [27], indicating that extra-framework Al species are marginally populated at the same time, suggesting that enhanced Lewis acidity that was observed in the IR spectra stems from Cu species.

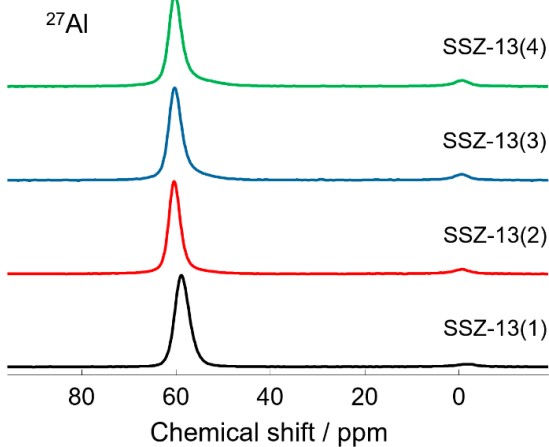

**Figure 3.** $^{27}Al$ MAS NMR spectra of SSZ-13.

Figure 4a displays the N$_2$ adsorption-desorption isotherms of the studied Cu-SSZ-13 materials. According to the IUPAC classification [28], all isotherms of the samples exhibit a mixed Type I and Type IVa isotherm with an H1 type hysteresis loop in the range of p p$_0^{-1}$ > 0.9, indicating the presence of the inter-crystalline meso-/macroporosity. The values of specific micropore volume are similar in all samples and are common for highly crystalline SSZ-13 material (Table 2). A general trend of an increase of the specific surface area and total pore volume in the samples prepared from the systems with a reduced TMAdaOH quantity (Cu-SSZ-13(2–4)) is observed (Table 2) in accordance with the SEM findings on the decreased crystal size (Figure 2).

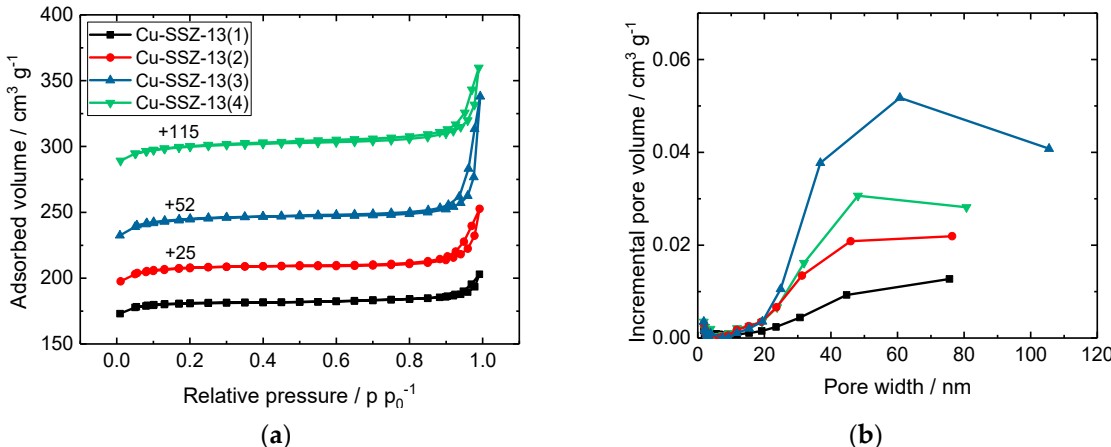

**Figure 4.** (**a**) N$_2$-sorption isotherms and (**b**) pore size distribution of Cu-SSZ-13.

The quantity of the copper introduced to the chabazite materials varies slightly in the range of 4.4–5.2 wt. % (Table 2). From this constant copper quantity, we can conclude on similar framework composition; thus, the comparable ion-exchange capacity (n(Cu)/n(Al) = 0.72–1.10) of the studied chabazites. In all the samples, the copper loading is higher than reported for Cu-SSZ-13 prepared by the Cu$^{2+}$ ion-exchange method (3.1–3.6 wt. %) [29,30].

**Table 3.** Acid site density determined in quantitative IR studies of NH$_3$ sorption ($C_{Brönsted}$ and $C_{Lewis}$), EPR relative intensities (Rel. Int.) of the fresh samples before dehydration and turnover frequency (TOF).

| Sample | $C_{Brönsted}$ /µmol g$^{-1}$ | $C_{Lewis}$ /µmol g$^{-1}$ | Rel. Int. | TOF (150 °C) 10$^3$/s$^{-1}$ |
|---|---|---|---|---|
| Cu-SSZ-13(1) | 180 | 1150 | 6.57 | 0.55 |
| Cu-SSZ-13(2) | 0 | 1395 | 1.00 | 0.52 |
| Cu-SSZ-13(3) | 20 | 880 | 1.27 | 0.46 |
| Cu-SSZ-13(4) | 35 | 825 | 1.69 | 0.42 |

## 2.2. Status of Copper Oxide Species

The copper speciation was broadly evaluated by DR UV-Vis, IR and EPR spectroscopy as well as TPR-H$_2$ studies to show evidence for the difference in the properties of the Cu moieties dispersed in zeolites SSZ-13(1–4). Although copper ions were introduced in the form of divalent cations, part of Cu still could undergo reduction during calcination and the presence of Cu$^+$ is definitely supposed. Moreover, both vacuum pretreatment (FT-IR experiments) and He-treatment applied before catalytic tests also provoke partial reduction of Cu$^{2+}$ species to Cu$^+$. A first insight into the speciation of copper sites can be provided by DR UV-Vis analysis (Figure 5). All investigated materials have two distinct bands in DR UV-Vis spectra. The charge transfer band at around 211–215 nm is related to O → Cu charge transition (CT) from lattice oxygen to isolated Cu$^+$/Cu$^{2+}$ species stabilized by the zeolite framework [31,32]. The broad absorption band between 500 and 800 nm is assigned to the

d → d transition of $Cu^{2+}$ [32,33]. An additional band at around 245 nm in the spectra of Cu-SSZ-13 is attributed to CuO species [34], while the band at ca. 325 nm proves the presence of oligomeric $Cu^{2+}$–$O^{2-}$–$Cu^{2+}$ chains [35]. Based on the DR UV-Vis spectra analysis, it can be concluded that copper species were introduced to the Cu-SSZ-13(2–4) materials in a similar aggregation state. On the contrary, the spectra of Cu-SSZ-13(1) revealed mainly the contribution of monomeric cations stabilized by the chabazite framework in this sample. Similar results were provided by ammonia sorption followed by IR spectroscopy.

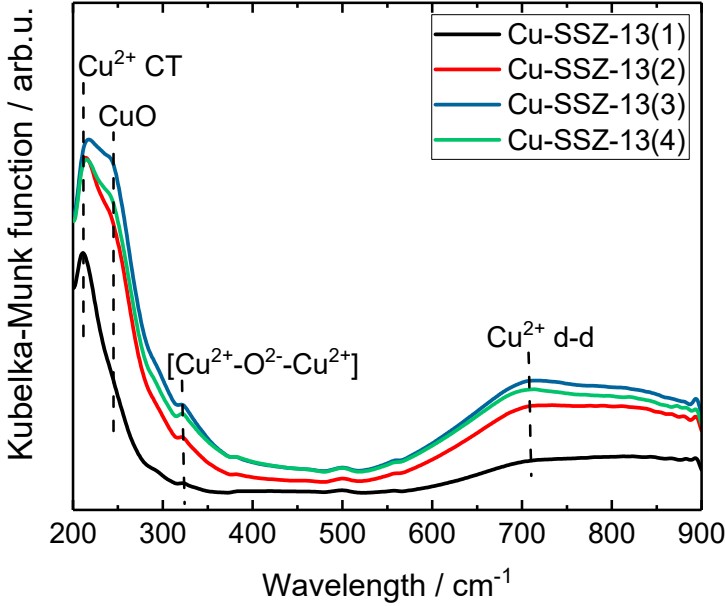

**Figure 5.** DR UV-Vis spectra of the Cu-exchanged chabazite samples.

The reducibility of the Cu-SSZ-13 materials was confirmed by the TPR experiments (Figure 6). Although it is difficult to differentiate the nature of copper species located in different sites, the identification of Cu species could be attained by comparing TPR-$H_2$ with the literature data. All the Cu-containing chabazite samples exhibit three-reduction peaks in the temperature range of 50–800 °C. The low-temperature peaks (<350 °C) were assigned to the reduction of isolated $Cu^{2+}$ in cha cage to $Cu^+$ ions ($Cu^{2+} + 1/2\,H_2 \rightarrow Cu^+ + H^+$) and to the reduction of CuO to $Cu^0$ ($Cu^{2+} + H_2 \rightarrow Cu^0 + 2\,H^+$). Namely, the $Cu^{2+}$ ions located in the pore openings, i.e., inside the cha cages next to eight-membered rings are expected to be more easily accessed by $H_2$. The reduction peak in the range of 350–600 °C is attributed to the reduction of $Cu^{2+}$ in the six-membered ring to $Cu^+$ ($Cu^{2+} + 1/2\,H_2 \rightarrow Cu^+ + H^+$), while the high-temperature peaks above 600 °C were associated with the reduction of highly stable $Cu^+$ ions to $Cu^0$ ($Cu^+ + 1/2\,H_2 \rightarrow Cu^0 + H^+$) [36,37]. The assignment of the dimeric Cu centers, i.e., $[Cu\text{-}O\text{-}Cu]^{2+}$ is less intensively investigated in the literature. The reduction of $[Cu–O–Cu]^{2+}$ also occurs in two stages as isolated monomeric $Cu^{2+}$ ions [38]. Moreover, Verma et al. [39] claimed that the $[Cu–O–Cu]^{2+}$ preferred coordination in eight-membered rings. Thus, in our case, such species appear to be reduced below <350 °C. The amount of $H_2$ consumed during TPR-$H_2$ provides qualitative information on the number of reducible Cu ions in the studied samples (Table 2). Verma et al. [39] also evaluated the Cu species in Cu-SSZ-13 catalysts with an n(Si)/n(Al) atomic ratio of 4.5 and found that there is a limit for the density (n(Cu)/n(Al) = 0.2) of the isolated $Cu^{2+}$ at the six-membered rings in SSZ-13. Beyond this limit, a part of the isolated $Cu^{2+}$ ions converts to Cu dimers. In our case, all Cu-SSZ-13(1–4) materials revealed n(Cu)/n(Al) in the range of 0.72–1.10 which indicates the formation of Cu dimers of oxide nature, in line with DR UV-Vis spectra.

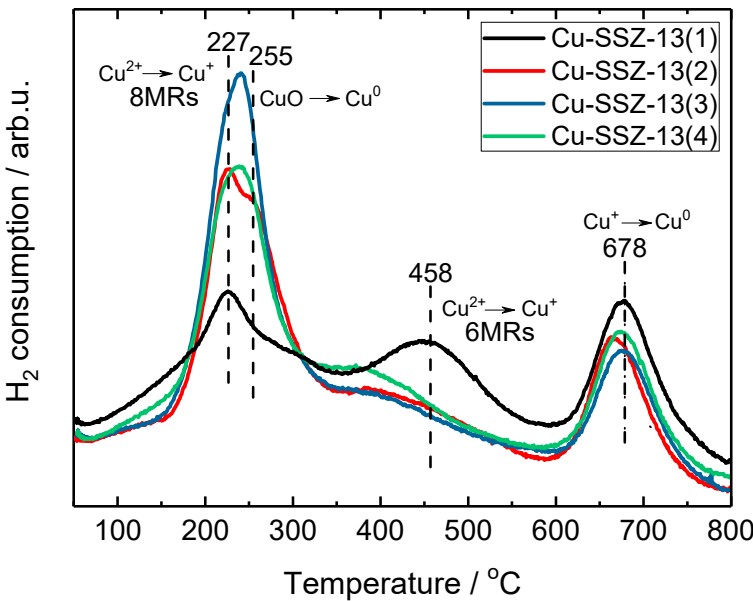

**Figure 6.** TPR profiles of the Cu-exchanged chabazite materials.

To further elucidate the presence of various copper speciation forms in Cu-SSZ-13(1–4), IR studies of CO sorption were carried out. At room temperature, the CO molecule is able to interact with exchangeable $Cu^+$ to form monocarbonyls (2155 $cm^{-1}$) or dicarbonyls (2150 and 2178 $cm^{-1}$) and $Cu^+$ in oxide-like phase (2140 $cm^{-1}$). Stable carbonyls formed with the participation of $Cu^{2+}$ ions can be observed only at temperatures as low as −100 °C [40–42]. The sorption of CO at −130 °C on the all samples studied reveals the bands (2250–2190 $cm^{-1}$) originating from $Cu^{2+}$ carbonyl species in Cu-SSZ-13(2) only (Figure 7a). The monocarbonyl $Cu^{2+}_{oxo}(CO)$ band (ca. 2210 $cm^{-1}$) is characterized by the very low value of the absorption coefficient. Moreover, the intensity of 2210 $cm^{-1}$ band can be influenced by neighboring $Cu^+$ carbonyl originated bands. Finally, the amount of $Cu^{2+}$ species cannot be easily estimated from the low-temperature IR spectrum. Such species are, however, undoubtedly present in the zeolites studied. If they cannot be detected with CO in other catalysts studied, it can be supposed that the electron-acceptor properties of $Cu^{2+}_{oxo}$ species are not sufficiently strong to effectively bond CO molecules.

The spectra of CO adsorbed at room temperature (RT) on Cu-SSZ-13 up to maximum intensities of the $Cu^+(CO)$ and $Cu^+_{oxo}(CO)$ monocarbonyl bands are presented in Figure 7b. The $Cu^+(CO)$ monocarbonyl bands in chabazite are shifted towards lower frequencies (2155 $cm^{-1}$) compared to ZSM-5 (n(Si)/n(Al) = 27; 2158 $cm^{-1}$) and have a broader half-width [43]. This indicates a stronger neutralization of $Cu^+$ by oxygen atoms of the chabazite framework, which is related to the higher abundance of aluminium atoms in chabazite than in ZSM-5. In turn, the broader half-width of $Cu^+(CO)$ band indicates the high heterogeneity of $Cu^+$ sites, i.e., the presence of $Cu^+$ exchangeable cations located in various position in the CHA framework, therefore, offering differentiated electron-donor properties. The maximum intensity of the moncarbonyls bands, both $Cu^+(CO)$ and $Cu^+_{oxo}(CO)$ can be taken as the measure of the $Cu^+$ ions concentration in both forms. From the spectra of CO interacting with redox sites in chabazites, we can confirm the presence of exchangeable copper cations (2156 $cm^{-1}$) as the main species. The only exception is the Cu-SSZ-13(2) material accommodating also copper oxide $Cu^+$ forms, identified by the presence of the $Cu^+_{oxo}(CO)$ band - 2140 $cm^{-1}$ ($Cu^+$ from oxide-like phase).

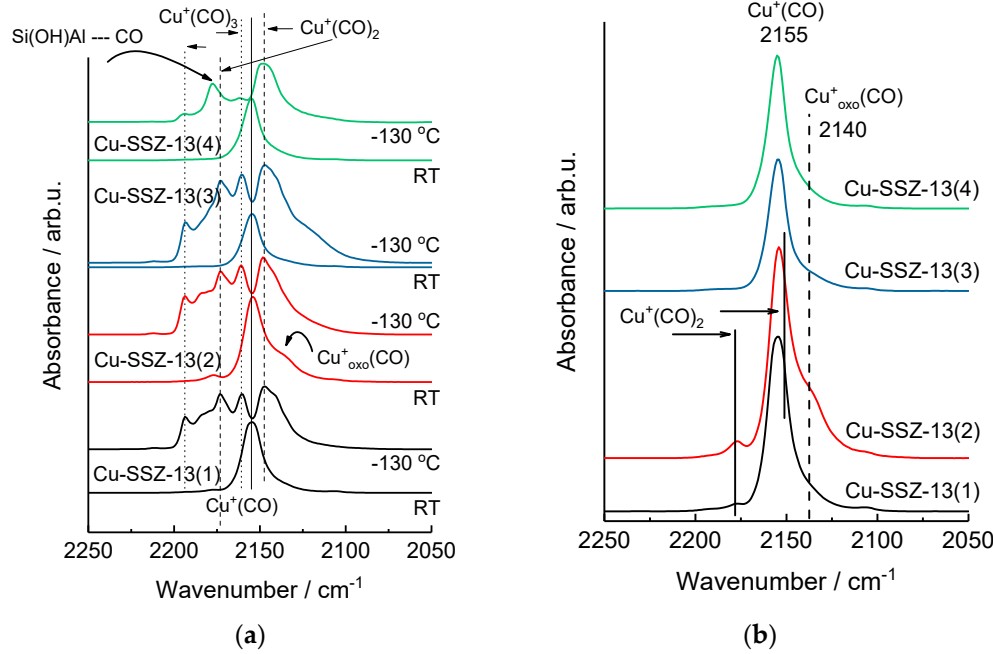

**Figure 7.** (**a**) The FTIR transmission spectra of CO adsorbed at −130 °C and room temperature (RT) in the Cu-SSZ-13 zeolites. (**b**) The maximum intensity of the $Cu^+(CO)$ and $Cu^+_{oxo}(CO)$ monocarbonyl bands (RT).

At room temperature, the mono $Cu^+(CO)$ and dicarbonyl $Cu^+(CO)_2$ complexes are stable, which emerge to tricarbonyl $Cu^+(CO)_3$ adducts at −130 °C. For all the samples studied, the dicarbonyl species are still detected at −130 °C; they are not transformed in tricarbonyl moieties. This evidences that some fraction of $Cu^+$ is the least accessible or cannot be withdrawn from these more shielded position sites in spite of the solvatation by CO molecules. Still, two coordination of $Cu^+$ are accessible to reagent molecules as $NH_3$ and NO in $NH_3$-SCR; thus, all the $Cu^+$ cations can be considered as providing catalytic activity in real process.

EPR allows identifying the coordination environment of isolated $Cu^{2+}$ ions because all the other Cu species ([Cu–O–Cu]$^{2+}$ or $Cu^+$) are EPR silent [30,44]. Figure 8a shows very similar EPR spectra characterized by a single isotropic peak centered at $g_{iso} \approx 2.17$, comparable to the one observed for copper in aqueous solutions, in agreement with previous studies [45]. In the hydrated zeolites (i.e., calcined in the air), the presence of water molecules in the coordination sphere of $Cu^{2+}$ ions leads to their detachment from the framework. As a result, the EPR spectra show an overlap of isotropic and anisotropic features. The former is due to aqueous complexes, which tend to move towards large cavities of the zeolite where they can freely rotate on the timescale of an EPR experiment: in this case, the hyperfine and *g* tensors are averaged. The latter comes from Cu ions attached to the framework and, thus, is responsible for the anisotropic components. The relative intensities of the samples were calculated as the double integral of the RT signal and normalized with respect to the weight of each measured sample. The values changed from 6.57 for Cu-SSZ-13(1) to 1.00 for Cu-SSZ-13(2). For Cu-SSZ-13(2) and Cu-SSZ-13(4), the relative intensity reached 1.27 for Cu-SSZ-13(3) and 1.69 for Cu-SSZ-13(4), respectively (Table 3).

After the samples dehydration, a well-defined parallel hyperfine structure of $Cu^{2+}$ appeared for all the spectra (Figure 8b), although the intensity of the signal decreased about 70% compared to the intensity of the initial zeolites. These results are in line with previous investigations over Cu-containing zeolites ZSM-5 and Y [46,47]. The $Cu^{2+}$ cations interact with the framework oxygen atoms more strongly than with oxygen atoms in water molecules. In other words, the positive charge of $Cu^+$ cations is more effectively balanced by the zeolite framework than by water molecules from the coordination sphere.

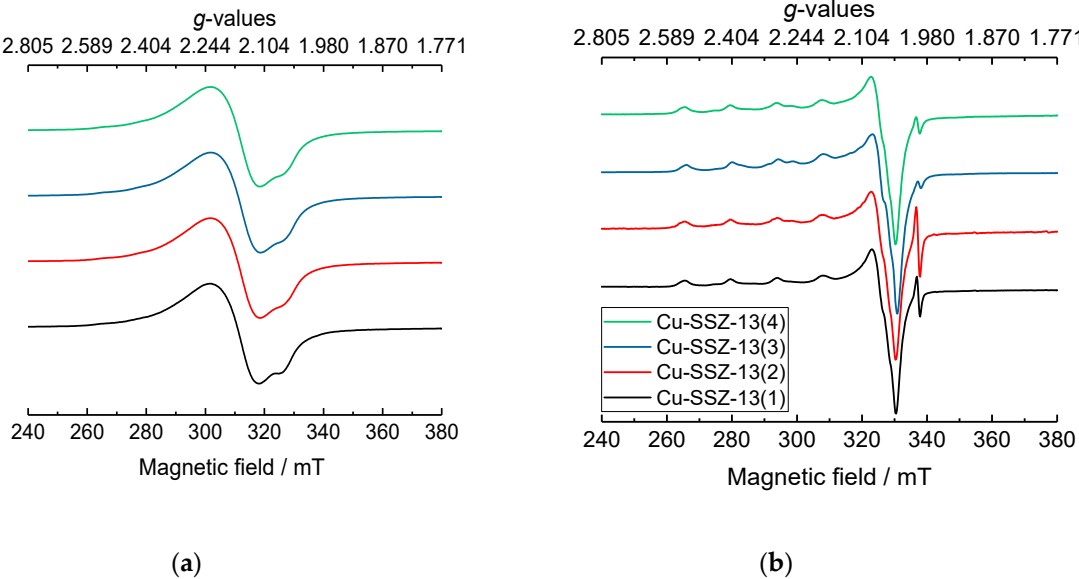

**Figure 8.** (**a**) EPR spectra of Cu-SSZ-13 recorded at RT and (**b**) EPR spectra of the activated Cu-SSZ-13; activation at 400 °C for 2 h reaching a final pressure of $10^{-3}$ mbar.

The presence of more than one Cu species can be also deduced from the EPR signals (Figure 8). Upon hydration sphere removal, the isolated $Cu^{2+}$ species migrate within the extra-framework positions, therefore, the dehydrated samples are enriched in Cu species in comparison with the fresh, hydrated material [45]. In particular, three different Cu sites (namely *A*, *B* and *C*) were identified by the spectral simulations illustrated in Figure 9 (Table 4 for the spin Hamiltonian parameters). The distribution of the sites is different for the four Cu-SSZ-13 samples (Table 5) and is given by the weight with which the simulations were performed. The EPR parameters of *A* and *B* are very similar to the ones found for the Cu-CHA zeolites in earlier studies [48,49]. They are usually associated with $Cu^{2+}$ ions strongly bound to the oxygen of the framework in six-membered ring (6MRs) sites with two aluminum atoms that compensate the copper charge.

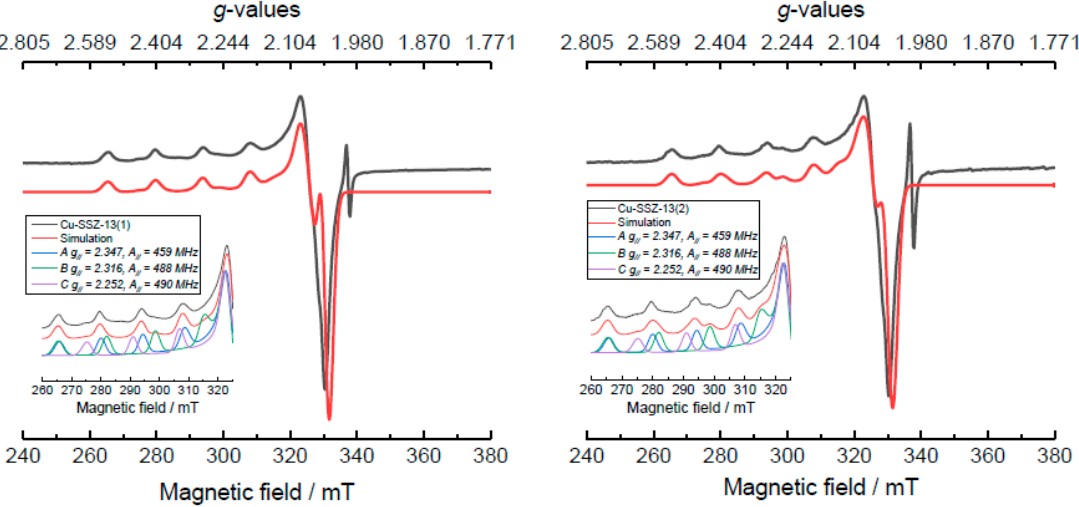

**Figure 9.** *Cont.*

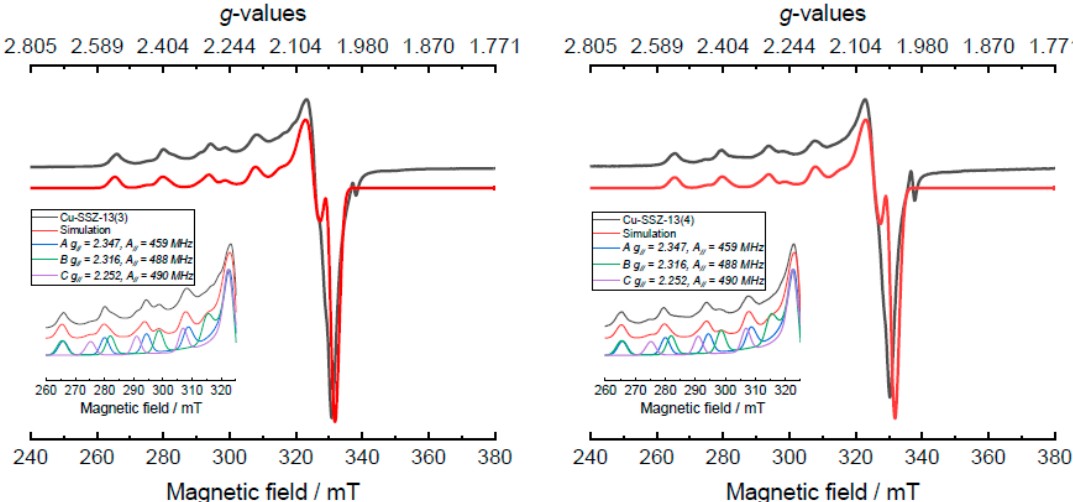

**Figure 9.** Experimental (black) and simulated (red) spectra of the activated Cu-SSZ-13(1–4). The low field part of the experimental and simulated spectrum is reported on the bottom left of each spectrum. The simulation is obtained by summing the spectra of three different Cu species (*A* in blue, *B* in green and *C* in violet) with a proper weight.

**Table 4.** Spin Hamiltonian parameters and broadening values (lwpp) used for the simulations of Cu species in the activated Cu-SSZ-13(1–4) samples.

| Cu Site | $g_\perp$ | $g_{//}$ | $A_\perp$/MHz | $A_{//}$/MHz | lwpp/mT |
|---------|-----------|----------|---------------|--------------|---------|
| A | 2.055 ± 0.003 | 2.347 ± 0.002 | 45 ± 8 | 459 ± 10 | 2.8 ± 0.3 |
| B | 2.058 ± 0.003 | 2.316 ± 0.004 | 38 ± 8 | 488 ± 7 | 2.8 ± 0.3 |
| C | 2.063 ± 0.003 | 2.252 ± 0.003 | 44 ± 5 | 490 ± 8 | 2.8 ± 0.3 |

**Table 5.** Percentage composition of the different Cu species used for the spectra simulations of the activated Cu-SSZ-13(1–4) samples.*

| Sample | *A*/% | *B*/% | *C*/% |
|--------|-------|-------|-------|
| Cu-SSZ-13(1) | 80 | 10 | 10 |
| Cu-SSZ-13(2) | 64 | 24 | 12 |
| Cu-SSZ-13(3) | 64 | 24 | 12 |
| Cu-SSZ-13(4) | 70 | 15 | 15 |

\* An uncertainty of 2 % has been estimated during the spectral simulations.

Species *A* shows a higher $g_{//}$ component and a lower $A_{//}$ parameter with respect to *B* (2.347 and 459 MHz for *A* versus 2.316 and 488 MHz for *B*). This variation in the two spin Hamiltonian models can be explained by considering essentially two factors: the coordination environment and the charge at the Cu centers. A decrease of the parallel component of the *g* tensor and an increase of the $A_{//}$ could be linked to a decrease of the coordination number and a change of the local geometry of the copper ions [50]. However, also the formal charge experienced by the cupric ions plays a remarkable role on the correlation between $A_{//}$ and $g_{//}$: in particular, it was proven that an increase of the negative charge or the delocalization of the charge on the Cu center corresponds to an increase in $A_{//}$ and a decrease in $g_{//}$ [51]. Thus, the difference in the EPR values for sites *A* and *B* could be caused by different charges at the Cu centers as well as the coordination number. Regarding the *C* species, their assignment is more challenging: the $g_{//}$ value is too low in respect to the one possessed by the copper in 6MRs. Moreover, these sites represent the 10%–15% of the total signal obtained from the simulations. Kwak et al. [12] suggested that the 6MRs sites are preferred at low copper loading, whereas other sites, likely located in the 8MRs can be populated only at high copper content. Since the four samples are characterized by a high amount of Cu, the chance to have Cu ions also in other sites is not negligible. According to literature reports [12] and the TPR analysis, we assigned the signal from *C* to cupric ions located

in 8MRs. Finally, the difference among the four samples synthesized depends on the distribution of the three species: in particular, Cu-SSZ-13(1) and Cu-SSZ-13(4) have a higher amount of *A* sites with respect to the other two in which the amount of *B* species is slightly larger.

## 2.3. Catalytic SCR Studies

Figure 10 shows the comparison of NO conversion and $N_2O$ yield over the studied Cu-SSZ-13 catalysts. The activity (Figure 10a) is almost the same for all catalysts up to 350 °C, but there is a large difference in activity of the catalyst above this temperature. The activity of the Cu-SSZ-13(1) is higher than that of Cu-SSZ-13(2–4) at >350 °C. NO conversion decreased for Cu-SSZ-13(2–4) due to side-reaction of $NH_3$ oxidation. The amount of $N_2O$ formed did not exceed 55 ppm in the whole studied temperature range. Similarly to the activity order, we observe the lowest yield of $N_2O$ for Cu-SSZ-13(1). Turnover frequency (TOF, Table 3) was defined as the moles of NO molecules converted over the individual Cu site per second in $NH_3$-SCR. We assumed that each Cu atom (ICP-OES analysis) acts as an active center. Clearly, we observed a decrease in TOF values form $0.55 \times 10^3$ s$^{-1}$ for Cu-SSZ-13(1) to $0.42–0.52 \times 10^3$ s$^{-1}$ for Cu-SSZ-13(2–4) prepared with lower quantity of TMAdaOH. Since the Cu content in all catalysts is almost the same and varied in the range of 4.4–5.2 wt. %, while the highest activity is achieved by materials with the lowest copper loading (i.e., 4.4 wt. %), we aimed to understand the nature of Cu species in Cu-SSZ-13 catalysts. TPR and EPR revealed more $Cu^{2+}$ located in the six-membered ring of Cu-SSZ-13(1) and less CuO species, which lead to its higher operating temperature up to 450 °C compared to other materials. Furthermore, the Cu-SSZ-13(2–4) catalysts show a high amount of CuO species. It is well known that these species are the copper active species for $NH_3$ oxidation at high temperature [52]. The accurate quantification of the content of $Cu^{2+}$ located in different sites of the CHA structure is within the scope of such studies. The catalysts with stronger Brønsted acid sites, i.e., isolated Al atoms, were found to exhibit high selectivity to nitrogen, especially in the high temperature range. The ammonia chemisorbed on the Brønsted acid sites is protected against oxidation and thus possesses ability to reduce the nitrogen oxides [53]. Together these factors determine the higher activity of Cu-SSZ-13(1) and lower yield of $N_2O$.

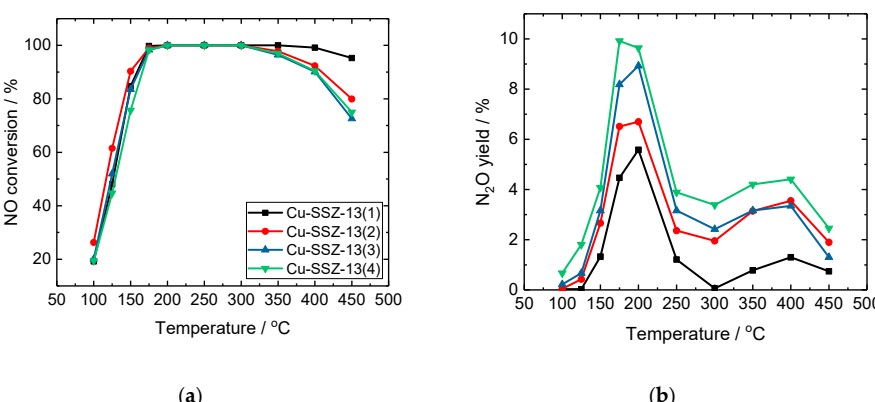

(a) (b)

**Figure 10.** (a) NO conversion and (b) $N_2O$ yield during SCR over the Cu-SSZ-13 catalysts (reaction conditions: 0.1 g of catalyst, 60 mL min$^{-1}$, 500 ppm NO, 575 ppm $NH_3$, 4 Vol. % $O_2$ and He balance, GHSV = 60 000 h$^{-1}$).

The comparative studies of $NH_3$-SCR over Cu-containing zeolites with different topologies revealed that the cage-type zeolite catalysts (e.g., Cu-SSZ-13, Cu-SSZ-16, Cu-SSZ-17) showed superior SCR activity as compared to the straight-channel frameworks (e.g., Cu-ZSM-5, Cu-Beta, Cu-Y) and hybrid types (e.g., Cu-ZSM-34, Cu-UZM-12) [54,55]. Thus, in Table 6, we compare the studied Cu-SSZ-13 with those reported in the scientific literature (concerning only Cu-containing SSZ-13) in terms of operation temperature for achieving > 80% of NO conversion. The formation on products ($N_2$, $N_2O$ and $NO_2$) was also provided.

**Table 6.** Comparison of the catalytic activity of Cu-SSZ-13 catalysts prepared with those reported in the literature.

| Sample | Reaction Conditions | Operation Temperature for Achieving > 80% NO Conversion | Formation of By-Products | Ref. |
|---|---|---|---|---|
| Cu-SSZ-13(1) Cu-SSZ-13(2–4) Ion-exchange; 4.4–5.3 wt. % of Cu | 500 ppm NO, 575 ppm $NH_3$, 4 Vol. % $O_2$ and balance He, GHSV = 60,000 $h^{-1}$ | 175–450 °C<br><br>175–400 °C | <6% $N_2O$ yield; <45 ppm $N_2O$<br><br><10% $N_2O$ yield; <55 ppm $N_2O$ | This work |
| Cu-SSZ-13 Ion-exchange; 2.9 wt. % of Cu | 500 ppm NO, 500 ppm $NH_3$, 10 Vol.-% $O_2$ and balance $N_2$, GHSV not provided | 200–500 °C | not shown | [56] |
| Cu-SSZ-13 Ion-exchange; 2.2–2.8 wt. % of Cu | 500 ppm NO, 500 ppm $NH_3$, 5 Vol.-% $O_2$, 5 Vol. % $H_2O$ and balance $N_2$, GHSV = 400,000 $h^{-1}$ | 225–525 °C | <10 ppm $N_2O$ | [57] |
| Cu-SSZ-13 Ion-exchange; 4.76 wt. % of Cu | 1000 ppm NO, 1000 ppm $NH_3$, 6 Vol. % $O_2$, 5 Vol. % $H_2O$ and balance He, GHSV = 300,000 $h^{-1}$ | 175–500 °C | <3% $N_2O$ yield | [54] |
| Cu-SSZ-13 Ion-exchange; 4.0–4.9 wt. % of Cu | 500 ppm NO, 500 ppm $NH_3$, 10 Vol. % $O_2$ and balance $N_2$, GHSV = 30,000 $h^{-1}$ | 200–500 °C | <25 ppm $N_2O$ | [58] |
| Cu-SSZ-13 Ion-exchange; 3.97 wt. % of Cu | 500 ppm NO, 500 ppm $NH_3$, 10 Vol. % $O_2$ and balance $N_2$, GHSV = 100,000 $h^{-1}$ | 175–450 °C | >95% $N_2$ selectivity | [59] |
| Cu-SSZ-13 Ion-exchange; 3.97 wt. % of Cu | 300 ppm NO, 300 ppm $NH_3$, 5 Vol. % $O_2$, 3 Vol. % $H_2O$ and balance $N_2$, GHSV not shown | 200–550 °C | not shown | [60] |
| Cu-SSZ-13 Ion-exchange; 1.00–1.17 wt. % of Cu | 500 ppm NO, 500 ppm $NH_3$, 10 Vol. % $O_2$, 3 Vol. % $H_2O$ and balance $N_2$, GHSV = 40,000 $h^{-1}$ | 250–550 °C | <6 ppm $N_2O$ | [17] |
| Cu-SSZ-13 Ion-exchange; 1.45–1.62 wt. % of Cu | 350 ppm NO, 350 ppm $NH_3$, 14 Vol. % $O_2$, 2.5 Vol. % $H_2O$ and balance $N_2$, GHSV = 200,000 $h^{-1}$ | 200–500 °C | not shown | [19] |

**Table 6.** *Cont.*

| Sample | Reaction Conditions | Operation Temperature for Achieving > 80% NO Conversion | Formation of By-Products | Ref. |
|---|---|---|---|---|
| Cu-SSZ-13 Hydrothermal synthesis; 2.5 wt. % of Cu | 500 ppm NO, 500 ppm $NH_3$, 5 Vol. % $O_2$ and balance Ar, GHSV = 180,000 $h^{-1}$ | 200–600 °C | not shown | [36] |
| Cu-SSZ-13 One-pot synthesis; 3.8 wt. % of Cu | 500 ppm NO, 500 ppm $NH_3$, 5 Vol. % $O_2$ and balance $N_2$, GHSV = 200,000 $h^{-1}$ | 175–550 °C | >90% $N_2$ selectivity | [13] |
| Cu-SSZ-13 Ion-exchange; 3.1 wt. % of Cu monolith, aged | 400 ppm NO, 500 ppm $NH_3$, 8 Vol. % $O_2$, 5 Vol. % $H_2O$ and balance Ar, GHSV = 30,300 $h^{-1}$ | 200–600 °C | <10 ppm $N_2O$ | [30] |
| Cu-SSZ-13 Ion-exchange; Content of copper not shown | 350 ppm NO, 350 ppm $NH_3$, 14 Vol. % $O_2$, 2 Vol. % $H_2O$ and balance $N_2$, GHSV = 30,000 $h^{-1}$ | 200–550 °C | <30 ppm $NO_2$ <30 ppm $N_2O$ | [55] |
| Cu-SSZ-13 Ion-exchange; 1.2 wt. % of Cu | 1000 ppm NO, 1000 ppm $NH_3$, 10 Vol. % $O_2$, 5 Vol. % $H_2O$ and balance $N_2$, GHSV = 200,000 $h^{-1}$ | 300–550 °C | <10 ppm $N_2O$ | [61] |

It can be found that the NO conversion achieved in the low temperature range (175–200 °C) in our catalysts is similar to the one reported for Cu-SSZ-13 prepared by ion-exchange [59] as well as one-pot synthesized Cu-SSZ-13 [13]. However, in our case, the operation temperature range is narrower (450 for our materials *versus* 550 °C reported in [13]). In addition, other listed examples showed more than 80% NO conversion up to 600 °C (even in the presence of $H_2O$) [30,61]. Prodinger et al. [19] focused (among others) on the effect of the particle size of the Cu-SSZ-13 in $NH_3$-SCR. In their studies, below 400 °C, all catalysts revealed a similar activity. Above 400 °C, the activity decreased due to competitive ammonia oxidation. Interestingly, the material with a higher particle size (mean particle diameter of 1.3 μm, 1.47 wt. % of Cu) revealed a slightly lower activity for $NH_3$ oxidation than other catalysts (average particle sizes of 260 and 450 nm, 1.45–1.62 wt. % of Cu). A similar trend was observed in our studies.

Since our Cu-SSZ-13(2–4) catalysts exhibited a high activity for $NH_3$ oxidation at high temperature due to the presence of CuO species, our next goal will be the application of selected materials in the selective catalytic ammonia oxidation into nitrogen and water vapor ($NH_3$-SCO). This statement can by supported by the investigations of Guo et al. [62] and Yu et al. [63], who reported that $NH_3$ is oxidized to $NO_x$ by surface CuO, and subsequently, $NO_x$ are reduced to $N_2$ and $H_2O$ by unreacted $NH_3$ on isolated $Cu^{2+}$ sites of Cu/SSZ-13 and Cu/SAPO-34, respectively.

## 3. Materials and Methods

### 3.1. Catalyst Preparation

Chabazite-type zeolite samples (CHA) have been prepared using commercial FAU-type zeolite material CBV760 (Zeolyst, Delfzijl, The Netherlands), sodium hydroxide (NaOH, pellets, 97 wt. %, Acros organics, Geel, Belgium), N,N,N-trimethyl-1-adamantammonium hydroxide (TMAdaOH, 20 wt. % solution, Sachem, Austin, TA, USA) and doubly distilled water. The needed amounts of the chemicals were mixed in order to prepare the synthesis mixtures of molar oxide compositions presented in Table 1. Briefly, for the synthesis of SSZ-13(1) 0.168 g of NaOH was dissolved in 0.975 g of water followed by the addition of 4.349 g TMAdaOH and 1 g of zeolite Y. The mixtures were transferred into polyterafluoroethene (PTFE)-lined autoclaves and subjected to hydrothermal treatment at 150 °C for a period of time from 5 to 10 days. The final product was recovered by filtration and washed repeatedly until the pH value of the supernatant had reached 7. The obtained solid has been calcined at 550 °C for 4 h and was subsequently subjected to three consecutive ion-exchanges with 0.5 M aqueous solution of ammonium nitrate (VWR, Leuven, Belgium) at 80 °C for 1 h. The protonic zeolite forms were obtained by calcining the ammonium-exchanged CHA-type materials at 550 °C for 4 h in static air. Further, the samples were ion-exchanged with 0.05 M aqueous solution of copper(II) acetate (Alfa Aesar, Kandel, Germany) at room temperature for 24 h. Finally, the materials were calcined at 550 °C for 4 h.

### 3.2. Catalyst Characterization

The chabazite materials were characterized by powder X-ray diffraction (XRD diffractometer, Panalytical Aeris Research Edition, Malvern, UK) using Cu Kα radiation ($\lambda$ = 0.15418 nm) in the 2θ range from 5° to 50° with a step size of 0.02°. The Cu-exchanged chabazite materials were characterized by powder X-ray diffraction (XRD diffractometer, HUBER G670, Rimsting, Germany) using Cu Kα radiation ($\lambda$ = 0.15418 nm) in the 2θ range from 5° to 80° with a step size of 0.05°.

The morphology of the samples was observed with a scanning electron microscope (SEM) LEO Gemini 1530 SEM from Zeiss (Oberkochen, Germany) using an acceleration voltage of 10 kV and a working distance of 10 mm. SEM-EDX experiments were performed at 20 kV. The metal loading of the samples was determined by an elemental analysis via optical emission spectrometry with inductively coupled plasma (ICP-OES, Optima 8000, Perkin Elmer, Rodgau, Germany). The samples (ca. 20 mg) were dissolved in a mixture of 2 cm³ HF (47–51 wt. %, NORMATOM®, Leuven, Belgium), 3 cm³ of $HNO_3$ (69 wt. %, ROTIPURAN® Supra, Karlsruhe, Germany) and 3 cm³ of HCl

(35 wt. %, ROTIPURAN® Supra, Karlsruhe, Germany) in a sealed PTFE vessel at 1100 W for 60 min in in a microwave oven (Multiwave 3000 Anton Paar). The local aluminum environment in the Cu-CHA samples was investigated through solid-state NMR spectroscopy using a Bruker Avance 750 spectrometer (magnetic field 17.6 T, Rheinstetten, Germany) at a frequency of 195.06 MHz for $^{27}$Al. The $^{27}$Al experiments were recorded at a spinning frequency of 12 kHz and a recycle delay of 0.1 s. A 1-μs pulse was used that corresponds to about a 30° pulse angle.

Textural properties of the H-CHA samples were determined from the $N_2$-sorption at −196 °C. The experiments were performed using an ASAP 2010 from Micromeritics (Norcross, GA, USA). The total pore volume was taken from the point $p\, p_0^{-1} = 0.995$. The specific surface area was calculated using the Brunauer–Emmett–Teller (BET) method and the pore width distribution was obtained using the Barret–Joyner–Halenda (BJH) method.

The acid properties (nature, concentration and strength of acid sites) of the catalysts were studied by FT-IR spectroscopy. Prior to measurements, all samples were pressed into self-supporting wafers (ca. 5 mg$^{-1}$ cm$^2$) and in situ thermally treated in a home-made quartz IR cell at 550 °C under high vacuum (p = $10^{-5}$ bar) for 1 h. As ammonia is the substrate molecules in $NH_3$-SCR, the acidic feature assessment was done in quantitative IR experiments with the use of ammonia as a probe molecule. The measurements were realized by saturation of all acid sites in the catalysts with ammonia (POCh, Gliwice, Poland) at −130 °C. Subsequently, physisorbed ammonia molecules were removed by 20 min evacuation. The concentrations of both Brønsted and Lewis acid sites were calculated from the maximum intensities of the $NH_4^+$ and $NH_3$-L bands and corresponding values of the absorption coefficients [19]. Additionally, the redox copper sites speciation was assessed in quantitative IR studies of CO sorption. The sorption of CO (Linde Gas, Kraków, Poland, 99.5 Vol. %) was performed on the vacuum pre-treated samples (550 °C, 1 h, p = $10^{-5}$ bar). Spectra were recorded with a Bruker Tensor 27 spectrometer (Karlsruhe, Germany) equipped with an MCT detector. The spectral resolution was 2 cm$^{-1}$.

The species and aggregation state of Cu-exchanged chabazite materials were studied by diffuse reflectance UV-Vis (DR UV-Vis) spectroscopy at room temperature on a Perkin Elmer Lambda 650 S instrument (Rodgau, Germany) equipped with a 150 mm integrating sphere using spectralon® (PTFE, reflective value 99%, Rodgau, Germany) as a reference.

The reducibility of the Cu-exchanged materials was investigated using temperature-programmed reduction (TPR) with $H_2$. The TPR-$H_2$ experiments were carried out on an AMI-100 apparatus, manufactured by Altamira Instruments (Pittsburgh, PA, USA). Prior to reduction, approximately 25 mg of the sample was oxidized in flowing synthetic air (30 cm$^3$ min$^{-1}$). Then the sample was reduced with 5 Vol. % $H_2$ diluted in an argon stream in the temperature range between 313 and 923 K and a heating ramp of 5 K min$^{-1}$. Thermal conductivity detector (TCD) was used to measure the $H_2$ consumption. For every experiment, a pulse calibration of the TCD was performed.

The concentration, structure and coordination environment of isolated $Cu^{2+}$ ions were investigated using electron paramagnetic resonance spectroscopy (EPR). The EPR spectra were recorded at room temperature with a Bruker EMXmicro X-band spectrometer (Rheinstetten, Germany). The modulation amplitude and the microwave power were kept at 10 G and 2 mW, respectively. The spectra were simulated by using the Easyspin software (version 6.0.0-dev.20) [64]. An axial spin Hamiltonian model was used for the simulation of each spectrum with a Voigtian line shape and an equal line width for all the species. The EPR spectra were recorded at room temperature for all Cu-SSZ-13 samples also in their dehydrated form.

The dehydrated samples were obtained by placing about 3–20 mg of Cu-SSZ-13 into a 4-mm EPR quartz tube sealed to a vacuum valve which was connected to a vacuum line and dehydrated under dynamic vacuum at 400 °C for 2 h, reaching a final pressure of $10^{-3}$ mbar.

### 3.3. Catalyst Experiments

The catalytic experiments were carried out in a fixed-bed quartz tube reactor (inner diameter: 6 mm, length: 200 mm). For catalytic experiments, a particle size fraction in the range of 0.200–0.400 mm was used. Prior to each experiment, the catalysts (100 mg) were outgassed at 350 °C for 1.5 h under 50 mL min$^{-1}$ of He and then cooled down to 50 °C. After that, the simulated flue gas, with a total flow rate of 60 mL min$^{-1}$ composed of 500 ppm NO, 575 ppm NH$_3$ and 4 Vol. % O$_2$ and balance He, was switched on to pass through the catalyst bed. The gas hourly space velocity (GHSV) was ~ 60,000 h$^{-1}$. The reaction was carried out at atmospheric pressure and in a range of temperatures from 50 °C to 450 °C with an interval of 25–50 °C. At each temperature, a steady state was reached within 70 min before quantification of NO and N$_2$O concentration. The gas leaving the reactor was washed in a gas-washing bottle filled with concentrated phosphoric acid. A NO$_x$-converter was used to reduce NO$_2$ to NO, in order to measure the total concentration of NO$_x$. An analysis of the NO and N$_2$O was performed using a non-dispersive infrared sensor (NDIR) URAS 10E (Fa. Hartmann and Braun, Frankfurt a. M., Germany). The conversion of NO (X(NO)) was determined according to X(NO) = ((c(NO)$_{in}$ − c(NO)$_{out}$/c(NO)$_{in}$) × 100%, where: c(NO)$_{in}$ and c(NO)$_{out}$ are the concentration of NO in the inlet gas and the concentration of NO in the outlet gas, respectively. The yield of N$_2$O (Y(N$_2$O)) was calculated based on the following equation: Y(N$_2$O) = (2 × c(N$_2$O)/(c(NO)$_{in}$ + c(NH$_3$)$_{in}$)) × 100%, where: c(N$_2$O), c(NO)$_{in}$, c(NH$_3$)$_{in}$ are concentration of N$_2$O in the outlet gas, concentration of NO and NH$_3$ in the inlet gas, respectively. The experimental uncertainty of the calculated conversion was ± 2%, as indicated by repeated measurements of identical catalysts.

The turnover frequency (TOF) values for NH$_3$-SCR were calculated by the following equation: TOF (s$^{-1}$) = (c(NO) × X(NO) × F)/(m/m$_W$), where c(NO), X(NO) and F are concentration of NO (mol ml$^{-1}$), NO conversion (%), volumetric flow rate (ml s$^{-1}$), respectively. Moreover, m and m$_W$ are the copper loading (g) on the catalyst and the molecular weight (63.54 g mol$^{-1}$) of copper, respectively.

### 4. Conclusions

SSZ-13 samples with TMAdaOH were successfully synthesized and applied as Cu-exchanged SSZ-13 to study their activity in NH$_3$-SCR. The synthesis of nanosized SSZ-13 (ca. 216–288 nm) was attained even with reduced TMAdaOH content. The benefit of decreasing the size of zeolite crystals is in our studies larger external surface area. The size of the crystals affects also the efficiency of the ion-exchange process, i.e., the quantity of the initial cations, which are exchanged with copper cations since the surface area available for direct contact with the solution depends on the size. Thus, the content of introduced Cu in all of the studied materials ranges from 4.4 to 5.2 wt. %. We vary the NaOH content in the the synthesis mixtures with the purpose to potentially have different Cu-exchange capacity together with different distribution of Cu species. All Cu-SSZ-13 materials exhibit the characteristic CHA structure, and Cu species are classified to unstable Cu$^{2+}$ cations inside the large cages of the CHA framework, CuO species and stable Cu$^{2+}$ in the six-membered rings. Since Cu$^{2+}$ cations located in the six-membered ring of the CHA framework are the active sites of catalyst, more Cu$^{2+}$ in the six-membered ring of Cu-SSZ-13(1) and less CuO species lead to its higher activity in NH$_3$-SCR compared to other materials. Cu-SSZ-13(2–4), which possess higher amount of CuO species is, thus, considered as the efficient catalyst of the selective ammonia oxidation into nitrogen and water vapor (NH$_3$-SCO) in the following studies.

**Author Contributions:** Author Contributions: Conceptualization, M.J. and A.P. (Ana Palčić); methodology, M.J.; investigation, A.P. (Ana Palčić), P.C.B., K.P., M.B., K.G.-M., D.P., M.J.; data curation, A.P. (Ana Palčić), P.C.B., K.P., M.B., K.G.-M., D.P., M.J.; writing—original draft preparation, M.J.; writing—review and editing, A.P. (Ana Palčić), K.G.-M., D.P., A.P. (Andreas Pöppl), R.G., M.J. All authors have read and agreed to the published version of the manuscript.

**Funding:** This work has been supported in part by Croatian Science Foundation under the project UIP-2019-04-4977. A.P. thanks to the Deutscher Akademischer Austauschdienst—German Academic Exchange Service (DAAD) for the research fellowship in the scope of the program Research Stays for University Academics and Scientists 2018 (57381327). K.G.M. acknowledgements the financial support from the National Science Centre, Poland (Grant No. 2015/18/E/ST4/00191).

**Acknowledgments:** We acknowledge support from the German Research Foundation (DFG) and Leipzig University within the program of Open Access Publishing. This work is part of a project that has received funding from the European Union's Horizon 2020 research and innovation programme under the Marie Skłodowska-Curie Grant Agreement No. 813209 (PARACAT).

**Conflicts of Interest:** The authors declare no conflict of interest.

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
