# Peer review of "Nanosized Cu-SSZ-13 and Its Application in NH3-SCR"

_catalysts, doi:10.3390/catal10050506_

Round 1
Reviewer 1 Report
The work is focused on the synthesis of Cu/SSZ-13 materials using a reduced amount of TMAdaOH and their application on the SCR reaction. The data reported are consistent and discussed in a accurate way, even if the degree of novelty appears to be low. Unfortunately the literature background is lacking and both the preparation method of SSZ-13 and the presumed reduce amount of TMAdaOH are not well discussed and contextualized. In the literature it is possible to find protocols for the SSZ-13 preparation that use less amount of TMAdaOH. Finally, because the formation of CuO, the Cu/SSZ-13 catalysts prepared with a lower TMAdaOH amount Cu/SSZ-13 (2-4) revealed to be less active compared to the reference one (Cu/SSZ-13 (1)) in the SCR reaction. Maybe a different application will be more suitable.
For these reason I don’t recommend the publication of the paper in Catalysts journal.
- Abstract is not clear. Please rewrite it in a more clear way.
- Also the introduction can be improved (e.g.: “Their results confirmed that isolated Cu2+ species associated with the 6-membered ring window, and placed into the ellipsoidal cavity of SAPO-34 were the active sites for NH3-SCR.” and “The isolated Cu2+ sites located in the 6-membered rings of the Cu-CHA catalysts are recognized as the active sites responsible for their superior NH3-SCR activity [7,11,12].”)
- The cited literature is quite incomplete. Several works on the same topic are missing. Please update it.
- Is the Cu-SSZ-13(1) sample prepared with the standard amount of TMAdaOH? Is the reported one the standard preparation protocol of SSZ-13? Searching in the literature I found different preparation procedures in which use lower TMAdaOH amount (e.g. Catalysis Today, 2015, 258, 347-358; Journal of Catalysis 331 (2015) 25–38 – SiO2 = 100; TMAdaOH = 10). Here the amount of TMAdaOH is higher (SiO2 = 100; TMAdaOH = 25/12.5)…. The different preparation techniques reported into the literature should be discussed in details and compared with the present one, to show the advantages of the latter over the others, eventually.
- Lines 80-81: “therefore can result in either high dispersion of Cu2O and CuO or the latter phases are of amorphous nature.”. Maybe speaking of copper dispersion in more correct.
- Lines 90-92: why sodium leads to more spherical particles and longer reaction times?
- Lines 93-94: the sentence is not clear.
- Tables start from table 2.
- Lines 162-163: “This can be explained by either the higher n(Cu)/n(Al) ratio of those zeolites or the various forms of copper sites; the latter is controlled by Al atoms location in the zeolite framework.” Why? Can you explain better? Please explain better the effect of Na+ atoms on aluminum sites and the Cu phase. By the contrast NMR of Al shows the same population of Al atoms. Maybe the acidity measurements of the bare materials, before the adding of Cu, cab be useful for the interpretation of the results. Note that also Lewis acidity of Cu-SSZ-13(3,4) is significantly less: this should be included in the discussion. Anyway oligomeric or bulky CuO species can (in agreement with the authors) the reason of the reduced acidity, even if small CuO particles can express Lewis acidity.
- DR-UV shows that Cu-SSZ-13(1) is the only sample where CuO is absent. It is correct? Can the authors explain this difference if Al atoms are similar in all samples, as well as the zeolite structure?
- About TPR, where the Cu-dimers reduce? In the same range of CuO? Lines 239-241: is the Cu-SSZ-13(1) containing the higher amount of isolated Cu2+ ions or the higher ratio (based on total Cu content)? If there is a limit that is determined by the Cu/Al ratio I suppose that the total amount of isolated Cu2+ sites should be more or less the same for all the samples (with minor differences derived by total Cu and Al content).
- About CO at -130 °C. What is the band around 2155-2160 cm-1?
- Lines 529-531: “Cu-SSZ-13(2-4), which possess higher amount of CuO species is, thus, considered as the efficient catalyst of the selective ammonia oxidation into nitrogen and water vapor (NH3-SCO) in the following studies.” I don’t agree with the conclusion since Cu-SSZ-13(1) is the best catalyst among the studied materials.
- In my view the main purpose of the work was not reached (“Low Cost Approach for the Preparation of Cu-SSZ-13 and Its Application in NH3-SCR”). Indeed the samples prepared with lower amount TMAdaOH of are less active.
Author Response
Dear Reviewer,
please find attached a reply.
Kind regards,
Magdalena Jablonska

Reviewer 2 Report
The manuscript entitled “Low Cost Approach for the Preparation of Cu-SSZ-13 and Its Application in NH3-SCR” by Palcic and co-workers is very complete and well executed. I recommend its publication in Catalysts.
Find enclosed some minor suggestions/questions/corrections:
Define SSZ-13 and SAPO-34 the first time it appears in the main text
The authors should expand the introduction with a broader vision of state of the art catalysts. The authors should also expand the SCR alternatives explored in the literature (with hydrocarbons, hydrogen, etc)
Include 3-D models to illustrate the different structures. Include a scheme of the synthesis and reactants.
Do the authors have any EDS analysis to complete the SEM characterization and evaluate the homogeneous distribution of elements?
Line 410: change more narrow by narrower
Table 6: Expand and discuss with other zeotypes and include selectivities to N2 and N2O
Table 1 should not be renamed according to its appearance in the main text?
Could the authors specify in more detail the treatment to prepare (digest) the samples for ICP-OES?
Author Response

(The authors gave the same response as above.)

Reviewer 3 Report
1. Why TMAdaOH was chosen as structure-directing agent? Cheaper agents are known, such as dimethylethylcyclohexylammonium halides [10.1016/j.cej.2018.10.053]. 2. Zeolite Y (Zeolyst CBV760) was used in the work as a source of aluminum and silicon . The SiO2 / Al2O3 molar ratio for this zeolite is 60, while in the present work the ratio 200 is stated. Please explain this. 3. The influence of the Si/TMAdaOH ratio has already been studied in some articles (eg, 10.1016/j.micromeso.2014.03.030). A low ratio (0.1, similar to the present work) has also been studied [10.1016/j.jcat.2012.10.029], and highly crystalline zeolites have been obtained in all cases. What is the novelty of this work compared to existing?4. “The sorption of CO at -130 °C on the all samples studied does not reveal any bands (2250-2190 cm-1) originating from Cu2+ carbonyl species (Figure 7a). Such species are however undoubtedly present in the zeolites studied. "
For samples 1–3, a band at ~ 2210 cm–1 is observed in the spectrum at -130 °С (Figure 7a). Can this band be attributed with Cu2+ carbonyl species? The authors do not discuss this band, although they discuss a band with comparable intensity in the spectra at room temperature.
5. Tab. 4. Perhaps the parameter A// for cites of type A is incorrect.
Author Response

(The authors gave the same response as above.)

Reviewer 4 Report
The presented work contains interesting findings but I would suggest the following modifications before the manuscript can be considered for the publication:
- The structure of the manuscript should be improved. Current version is similar to the report format rather than the publication. The results are well presented, however there is a lack of discussion, cross correlating the results obtained from different techniques. The comparison with the work already published should be improved.
- Line 26-27 - rewrite the sentence and the meaning is not very clear
- Introduction - there are lots of information anf finding on Cu-SSZ-13 structure prepared by others. However, the findings on NH3-SCR catalysis is missing. That should be included to justify the goal of this work. The authors should explain why they decided to focus on this reaction.
- Section 2 - Results and Discussion - before moving straight to XRD results there should be a paragraph on the studied catalytic systems.
- Section 2 - the information on the series of zeolites/catalysts (Table 1) should be included before any results are discussed.
- Table 2 & 3 Consider to present the data differently. It is not easy for the reader to look at two different column from two different tables.
- Line 95-96 - The paragraph only refers to Table 3.
- Line 170-171 - explain the shift for Cu-SSZ-13 (2-4)
- DR UV-Vis that should be defined at the beginning of the manuscript
- Line 257 - Since referred to ZSM-5 results please provide the information on the position of Cu+(CO) monocarbonyl bonds in ZSM-5
- Figure 7b - Is the data colllected at RT?
- Line 284-286 - Are the data on the relative intensity presented on the graph or table?
- Line 308 - can the parameters of g'' and A" be indicated on the graph
- Line 413 - the catalytic results only present NO conversion and N2O yield. Could the author add the data on NH3 oxidation since it is mentioned in the paragraph?
Author Response

(The authors gave the same response as above.)

Round 2
Reviewer 1 Report
The paper has been significantly improved that why I recommend its publication after minor revision.
- Lines 239-240: “An additional band at around 245 nm in the spectra of Cu-SSZ-13(2-4) is attributed to 239 CuO species [34]”. Concerning DR-UV spectra I agree on the possibility of CuO species over the sample 1. However the sentence in lines 239-240 seems to exclude this hypothesis.
- “Verma et al. [32] also evaluated the Cu species in Cu-SSZ-13 catalysts with n(Si)/n(Al) atomic ratio of 4.5 and found that there is a limit for the density (n(Cu)/n(Al) = 0.2) of the isolated Cu2+ at the 6-membered rings in SSZ-13. Beyond this limit, a part of the isolated Cu2+ ions converts to Cu dimers.” Following this paper, the maximum amount of isolated Cu2+ sites is limited by the Cu/Al ratio. Thus increasing the Al content also the number of isolated Cu2+ sites should increase. It follows that sample 1, that is the one with the higher Al, is the catalyst that can accommodate the larger amount of isolated Cu2+ (2.6 wt% / 27 g/mol * 0.2 – Verma ratio - = 0.019 molCu). The samples 2-4 can only accommodate less Cu2+ isolated sites because a lower Al content. However the sentence “In our case, Cu-SSZ-13(1) with the lowest n(Cu)/n(Al) = 0.72 possesses the highest amount of isolated Cu2+ at the 6-membered rings in SSZ-13 as revealed by TPR profile.” Is not completely correct. Indeed the same ratio can be obtained reducing both Cu and Al content for example by the half, but the maximum of isolated Cu atoms is only 0.0096 mol (following Verma).
Please correct me if I’m wrong.
Author Response
Dear Reviewer,
please find attached the revision.
Kind regards,
Magdalena Jablonska
